# Cannabidiol Induces Cell Cycle Arrest and Cell Apoptosis in Human Gastric Cancer SGC-7901 Cells

**DOI:** 10.3390/biom9080302

**Published:** 2019-07-25

**Authors:** Xin Zhang, Yao Qin, Zhaohai Pan, Minjing Li, Xiaona Liu, Xiaoyu Chen, Guiwu Qu, Ling Zhou, Maolei Xu, Qiusheng Zheng, Defang Li

**Affiliations:** 1Yantai Key Laboratory of Pharmacology of Traditional Chinese Medicine in Tumor Metabolism, School of Integrated Traditional Chinese and Western Medicine, Binzhou Medical University, Yantai 264003, China; 2School of Gerontology, Binzhou Medical University, Yantai 264003, China; 3School of Pharmacy, Binzhou Medical University, Yantai 264003, China

**Keywords:** Cannabidiol, gastric cancer, SGC-7901 cell, cell cycle arrest, apoptosis, ROS

## Abstract

The main chemical component of cannabis, cannabidiol (CBD), has been shown to have antitumor properties. The present study examined the in vitro effects of CBD on human gastric cancer SGC-7901 cells. We found that CBD significantly inhibited the proliferation and colony formation of SGC-7901 cells. Further investigation showed that CBD significantly upregulated ataxia telangiectasia-mutated gene (ATM) and p53 protein expression and downregulated p21 protein expression in SGC-7901 cells, which subsequently inhibited the levels of CDK2 and cyclin E, thereby resulting in cell cycle arrest at the G0–G1 phase. In addition, CBD significantly increased Bax expression levels, decreased Bcl-2 expression levels and mitochondrial membrane potential, and then upregulated the levels of cleaved caspase-3 and cleaved caspase-9, thereby inducing apoptosis in SGC-7901 cells. Finally, we found that intracellular reactive oxygen species (ROS) increased after CBD treatment. These results indicated that CBD could induce G0–G1 phase cell cycle arrest and apoptosis by increasing ROS production, leading to the inhibition of SGC-7901 cell proliferation, thereby suggesting that CBD may have therapeutic effects on gastric cancer.

## 1. Introduction

Gastric cancer is a common malignant tumor that originates in the gastric mucosal epithelium [1]. Gastric cancer can be further classified according to the disease site, which includes gastric cardia cancer, gastric cancer, and gastric antrum cancer [2]. Recent epidemiological data have indicated that there are about 951,000 new cases of gastric cancer around the world and about 723,000 deaths due to gastric cancer, thereby ranking gastric cancer as the fifth most common and third most malignant tumor with the highest mortality rate [3]. Countries including Japan, South Korea, and China are at high risk for gastric cancer [4]. Chinese gastric cancer morbidity and mortality account for 42.6% and 45.0% of global gastric cancer morbidity and mortality, respectively [5]. Gastric cancer is the most common malignant tumor in China [6], with regional differences in incidence. The incidences of gastric cancer in the northwest and eastern coastal areas of China are significantly higher than in the southern regions [7]. The high-risk age of the disease is above 50 years old, and the ratio of male to female risk is 2:1 [8]. In the past few decades, due to an increase in pressure at work and in daily life, dietary changes, and *Helicobacter pylori* infections, gastric cancer has been observed to have an earlier onset [9].

Current treatment schemes for gastric cancer mainly include surgery and chemotherapy: However, their long-term effects are not ideal [10]. For instance, tumor recurrence has been reported in half of the patients who undergo surgery [11]. In addition, chemotherapy, which induces tumor cell cytotoxicity and eventually death [12], has not significantly improved the overall survival of patients with gastric cancer because of poor selectivity and toxicity [2,10,13,14]. Thus, the traditional treatment schemes for gastric cancer do not meet clinical needs, and the development of novel treatment methods is imperative [15]. It is thus important to study the effect of traditional Chinese medicine on malignant tumors, to explore its mechanism of action, and to develop new and effective anti-tumor drugs [16].

Cannabidiol (CBD) is the main chemical component of the medicinal plant cannabis (*Cannabis sativa* L). This cannabinoid is extracted from female cannabis plants and is nonaddictive [17]. Studies have shown that CBD inhibits tumor cell proliferation, metastasis, or the induction of autophagy or apoptosis [18,19]. CBD and tamoxifen have been cocultured with C6 glioma cells, which showed an inhibitory effect on C6 glioma cells [20]. CBD induces apoptosis in human glioma cells U87 and U373 through mechanisms such as the activation of caspase and the involvement of reactive oxygen species (ROS) [21]. CBD induces mouse lymphoma EL-4 cells and Jurkat leukemia cell apoptosis by regulating NOX4 and p22phox expression, which in turn leads to an increase in reactive oxygen species (ROS) level [22]. CBD plays an antiprostate cancer role by inhibiting prostate cancer cell proliferation or inducing apoptosis [23]. CBD can also inhibit the growth and metastasis of breast cancer cells through the epidermal growth factor (EGF)/epidermal growth factor receptor (EGFR) [24] and protein kinase B (AKT)/mTOR/4EBP1 [25] signaling pathways. In addition, CBD exerts a good safety profile while exhibiting significant anticancer effects [19,26]. However, the effects of CBD on protein expression in gastric cancer cells and the underlying mechanism of action are unclear.

To explore the antitumor effects of CBD on gastric cancer, we selected human gastric cancer SGC-7901 cells as a research object. Preliminary experiments have shown that CBD can significantly inhibit the proliferation and induce apoptosis in SGC-7901 cells, suggesting that CBD may be a potential chemotherapeutic drug for gastric cancer. However, its specific mechanism of action is still unclear. In this study, we explored the in vitro effects of CBD on human gastric cancer SGC-7901 cells and its molecular mechanisms.

## 2. Methods

### 2.1. Cell Culture

Human gastric cancer SGC-7901 cells were obtained from Cell Bank, Typical Culture Preservation Commission, Chinese Academy of Sciences (Shanghai, China). The cells were cultured with RPMI 1640 (SH30809.01, GE Healthcare Life Sciences Hyclone Laboratories, Logan, UT, USA) containing 10% fetal calf serum (REF10091-48, Gibco, Invitrogen), 0.1 µg penicillin, and 0.1 µg/L streptomycin (P1400, Solarbio, Beijing, China) and were incubated in a 5% CO_2_ incubator (HF90/HF240, Heal Force, Shanghai, China) at 37 °C.

### 2.2. Cell Counting Kit-8 (CCK-8) Assay

The effect of CBD on the viability of SGC-7901 cells was determined using a CCK-8 assay. SGC-7901 cells were cultured in RPMI 1640 medium and, upon reaching the logarithmic growth phase, were digested with 0.25% trypsin + 0.02% ethylene diamine tetraacetic acid (EDTA), centrifuged at 600× *g* for 3 min, and collected. After counting, 100 µL of the cell suspension was seeded in each well of a 96-well plate at a density of 1.2 × 10^5^ cells/mL. After 24 h of culture, culture medium containing 5, 10, 20, 30, and 40 µg/mL of CBD was added and further cultured for another 24 h in the incubator. After 24 and 48 h of culture, 10 µL of CCK-8 solution was added to each well and incubated for 2–4 h. Then, the plate was shaken for 1 min in the dark, and the absorbance of each well at a wavelength of 450 nm was detected using a Thermo 3001 multifunction microplate (DMI3000, Leica, Germany) reader. The absorbance of the cells treated with 0.1% DMSO in RPMI 1640 was used as a control (survival rate: 100%). IC_50_ (half maximal inhibitory concentration) indicates a drug concentration resulting in a 50% reduction in cell survival.

### 2.3. Hoechst 33258 Staining

Approximately 2 mL of the SGC-7901 cell suspension was seeded into six-well plates at a density of 2.5 × 10^5^ cells/mL. After 24 h of culture, the CBD culture medium containing 10, 20, and 40 µg/mL was added, and the plate was incubated for 24 h. After incubation, the culture solution was removed by aspiration, and 0.5 mL of fixative was added to each well and incubated for at least 10 min (can be overnight at 4 °C). Then, the fixative was removed, and the cells were washed twice with phosphate buffered solution (PBS) for 3 min each time, with shaking. Approximately 0.5 mL of 2-µg/mL Hoechst 33258 was then added to stain the cells at room temperature for 5 min with shaking. To remove the staining solution, the cells were rinsed twice with PBS for 3 min each time, with shaking. A drop of a fluorescent antiquenching agent was placed on the slide, covered with a coverslip, and the cells were allowed to come into contact with the sealing solution, avoiding air bubbles. Fluorescence microscopy (DMI3000, Leica, Germany) could detect blue nuclei.

### 2.4. Colony Formation Assay

The SGC-7901 cells at the logarithmic growth phase were digested with 0.25% trypsin + 0.02% EDTA and centrifuged at 800× *g* for 3 min to collect the cells. After counting, the cell concentration was adjusted, and 2 mL of the cell suspension containing 300 cells was seeded into each well of a six-well plate. After 48 h of culture, the CBD culture medium containing 10, 20, and 40 µg/mL was added, and after culturing for another 24 h, the fresh medium was replaced, and cell growth was monitored each day. Cells were cultured for 14 days and then dyed. The medium was discarded, and the cells were washed twice with PBS. The cells were fixed with a 4% cell fixing solution for 15 min, and then the cell fixing solution was discarded. The cells were then stained with gentian violet for 10 min and then slowly rinsed with running water. The plates were then observed for the formation of colonies. Images were captured using a light microscope (XDS-3, OPTIKA, Ponteranica, BG, Italy). Clones containing more than 50 cells were counted and used in calculating the rate of colony formation.

### 2.5. Cell Cycle Analysis

The cell cycle distribution of CBD-treated SGC-7901 cells was determined by flow cytometry. The SGC-7901 cells were treated with different concentrations of CBD or treated with 24 h of CBD and subsequently incubated with fresh culture media for 24 h. Then the cells (approximately 1 × 10^6^ cells per well) were harvested and fixed overnight in 70% ethanol at 4 °C. After fixation, the cells were centrifuged at 3000× *g* for 5 min to remove the ethanol. Then, the cells were washed with PBS, treated with 100 µL of RNase A, resuspended, and incubated at 37 °C for 30 min in the dark. Fluorescence detection of propidium iodide (PI)-DNA complexes was determined by Epics XL flow cytometry (Beckman Coulter, Inc, Brea, CA, USA). Cell distribution in different stages of the cell cycle was analyzed using WinMDI 2.8 software (Scripps Research, La Jolla, CA, USA), and cell cycle distribution was calculated [27,28].

### 2.6. Annexin V-FITC/Propidium Iodide (PI) Double Staining Assay

The Annexin V-FITC/PI double staining assay was performed according to the manufacturer’s instructions. Briefly, SGC-7901 cells in a logarithmic growth phase were digested with 0.25% trypsin + 0.02% EDTA and centrifuged at 600× *g* for 3 min to collect cells. The cell density of the suspension was adjusted to 2.5 × 10^5^ cells/mL, and 2 mL of the suspension was loaded into each well. After 24 h of culture, CBD culture medium containing 10, 20, and 40 µg/mL was added, and the cultures were incubated for 24 h in the incubator. The cells were collected and centrifuged at 2000× *g* for 3 min at room temperature. The cells were resuspended in precooled 1× PBS and centrifuged at 2000× *g* for 3 min, and the cells were washed. The cells were resuspended by adding 500 µL of 1× binding buffer. Then, 5 µL of Annexin V-FITC was added to the suspension, mixed well, and incubated for 15 min at room temperature. The cells were then stained with 5 µL of a PI staining solution before loading into a flow cytometer. Fluorescence intensity was measured using a FACSCanto II flow cytometer (Becton Dickinson, Franklin Lakes, NJ, USA), and the apoptotic rates of CBD-treated cells were analyzed using FACSDiva software (version 6.1.3; Becton Dickinson, Franklin Lakes, NJ, USA).

### 2.7. Mitochondrial Membrane Potential Assay

The JC-1 method was performed according to the manufacturer′s instructions. Briefly, SGC-7901 cells in a logarithmic growth phase were digested with 0.25% trypsin + 0.02% EDTA and centrifuged at 600× *g* for 3 min to collect cells. The cell density of the suspension was adjusted to 2.5 × 10^5^ cells/mL, and 2 mL of the suspension was added to each well of a six-well plate. After 24 h of culture, CBD medium containing 10, 20, and 40 μg/mL was added, and the culture was cultured in an incubator for 24 h. Carbonyl cyanide 3-chlorophenylhydrazone (CCCP) (10 mM) was added to the cell culture medium at 1:1000 and diluted to 10 μM, and the positive control cells were treated for 20 min. The medium in the six-well plate was aspirated, the cells were washed once with PBS, 1 mL of the cell culture solution was added, 1 mL of JC-1 (CBIC2(3)) staining working solution was added, and the mixture was thoroughly mixed and incubated at 37 °C for 20 min. The supernatant was aspirated and washed twice with JC-1 staining buffer (1×), and 2 mL of the cell culture medium was added and observed under a fluorescence microscope.

On the other hand, the mitochondrial membrane potential of CBD-treated SGC-7901 cells was determined by flow cytometer. Briefly, SGC-7901 cells in a logarithmic growth phase were digested with 0.25% trypsin + 0.02% EDTA and centrifuged at 600× *g* for 3 min to collect cells. The cell density of the suspension was adjusted to 2.5 × 10^5^ cells/mL, and 2 mL of the suspension was added to each well of a six-well plate. After 24 h of culture, CBD medium containing 10, 20, and 40 μg/mL was added, and the culture was cultured in an incubator for 24 h. CCCP (10 mM) was added to the cell culture medium at 1:1000 and diluted to 10 μM, and the positive control cells were treated for 20 min. The medium was discarded and the cells were washed once with PBS. Then 1 mL of the cell culture media and 1 mL of JC-1 staining working solution was added and mixed well, and subsequently incubated at 37 °C for 20 min. After incubation, the cells were collected and centrifuged at 2000× *g* for 3 min at room temperature. The cells were resuspended in prechilled 1× JC-1 staining buffer, centrifuged at 2000× *g* for 3 min, and washed twice. The cells were resuspended by the addition of 500 μL of 1x JC-1 staining buffer and then loaded onto a flow cytometer. Fluorescence intensity was measured using a FACSCanto II flow cytometer (Becton Dickinson, Franklin Lakes, NJ, USA).

### 2.8. Reactive Oxygen Species Assay

The ROS method was measured according to the manufacturer′s instructions. Briefly, SGC-7901 cells in a logarithmic growth phase were digested with 0.25% trypsin + 0.02% EDTA and centrifuged at 600× *g* for 3 min to collect cells. The cell density of the suspension was adjusted to 2.5 × 10^5^ cells/mL, and 2 mL of the suspension was added to each well of a six-well plate. After 24 h of culture, CBD medium containing 10, 20, and 40 μg/mL was added, and the culture was cultured in an incubator for 24 h. Rosup (50 mg/mL) was added to the cell culture medium at 1:1000, and the positive control cells were treated for 20 min. The medium was aspirated in a six-well plate, the cells were washed once with PBS, 1 mL of 2′,7′-dichlorodihydrofluorescein diacetate (DCFH-DA) was added, and everything was mixed well and incubated at 37 °C for 20 min. The supernatant was aspirated and washed three times with serum-free medium, and 2 mL of serum-free medium was added and observed under a fluorescence microscope.

In addition, the ROS levels of CBD-treated SGC-7901 cells were determined by flow cytometer. Briefly, SGC-7901 cells in a logarithmic growth phase were digested with 0.25% trypsin + 0.02% EDTA and centrifuged at 600× *g* for 3 min to collect cells. The cell density of the suspension was adjusted to 2.5 × 10^5^ cells/mL, and 2 mL of the suspension was added to each well of a six-well plate. After 24 h of culture, CBD medium containing 10, 20, and 40 μg/mL was added, and the culture was cultured in an incubator for 24 h. Rosup (50 mg/mL) was added to the cell culture medium at 1:1000, and the positive control cells were treated for 20 min. The medium was aspirated in a six-well plate, the cells were washed once with PBS, 1 mL of DCFH-DA was added, and everything was mixed well and incubated at 37 °C for 20 min. The supernatant was aspirated, and the cells were collected and centrifuged at 2000× *g* for 3 min at room temperature and washed three times with serum-free medium. The cells were resuspended in serum-free medium and centrifuged at 2000× *g* for 3 min, and the cells were washed 3 times. The cells were resuspended by the addition of 500 μL of serum-free medium and then loaded onto a flow cytometer. Fluorescence intensity was measured using a FACSCanto II flow cytometer (Becton Dickinson, Franklin Lakes, NJ, USA).

### 2.9. Western Blotting Analysis

The SGC-7901 cells (density: 1 × 10^6^/well) were seeded into 100-mm culture dishes and treated with different concentrations of CBD for 24 h. After treatment, the cells were harvested and mixed with a cell lysis buffer (20 mM Tris-HCl, 150 mM NaCl, 1 mM EDTA, 1% Triton X-100, 2.5 mM sodium pyrophosphate, 0.1 mM phenylmethanesulfonylfluoride, 0.1 mM sodium orthovanadate, 0.5 mM dithiothreitol, 10× protease inhibitor; pH 6.8) on ice for 30 min. The suspension was then centrifuged at 12,000× *g* for 10 min at 4 °C. The supernatant was collected and stored at −20 °C. Protein concentration was determined using a bicinchoninic acid (BCA) protein assay kit (Solarbio, Beijing, China). A cell lysate having a protein content of 40 mg and an equal volume of sodium dodecyl sulfate (SDS) loading dye (2% SDS, 10% sucrose, 0.002% bromophenol blue, 5% 2-mercaptoethanol, 625 mM Tris; pH 6.8) was subsequently separated on ta 12.5% SDS-PAGE with an iridescent protein molecular marker (Solarbio, Beijing, China). After two hours of operation at 110 V, the proteins were transferred onto a polyvinylidene fluoride (PVDF) membrane (Millipore, REF 88520) using a Trans-Blot^®^ SD semidry transfer cell (Bio-Rad). The membrane was blocked with 5% bovine serum albumin (BSA) in Tris-buffered saline containing 0.1% Tween-20 (TBS-T) for 120 min, and then incubated overnight at 4 °C with primary antibodies (some were purchased from Abcam, Cambridge, UK; the others were purchased from cell signaling technology, Boston, United States) as follow: p21 (1:1000, ab109199), Bcl-xl/Bcl-2-associated death prompter (BAD, 1:2000, ab32445), cyclin-dependent kinase 2 (CDK2, 1:1000, ab32147), B-cell lymphoma-2 (Bcl-2, 1:500, ab196495), ataxia telangiectasia-mutated gene (ATM, 1:2000, ab199726), Cyclin E (1:5000, ab194070), Caspase-3 (1:1000, #9662), p53 (1:1000, #9284), apoptotic protease activating factor-1 (Apaf-1, 1:1000, #5088), cytochrome C (1:1000, #4280), Caspase-9 (1:1000, #9508), Bcl-2-associated X (BAX, 1:1000, #2772), active Caspase-3 (Cleaved Caspase-3, 1:1000, #9664), and active Caspase-9 (Cleaved Caspase-9, 1:1000, #52873)) containing 5% BSA. Subsequently, the membrane was washed with TBS-T buffer for 45 min and then hybridized with the appropriate horseradish-conjugated secondary antibody for 40 min. An electro-chemi-luminescence (ECL) chromogenic solution was added for color development, and the target gene was detected using an immunoblot imaging system. Immunoreactive bands were visualized using a Novex™ ECL chemiluminescent substrate reagent kit (WP20005; Thermo Fisher Scientific, Waltham, MA, USA) using a film processor (BioSpectrum Imaging System, Upland, CA, USA). Image-Pro Plus 6.0 (IPP6) software was used to calculate the gray-scale values of each band [29].

### 2.10. Statistical Analysis

The experiments were conducted at least three times. The data were presented as the mean ± standard deviation. Statistical analyses were performed using the SPSS 21.0 software package (version 21.0, SPSS Inc., Chicago, IL, USA). Student’s *t*-test or one-way ANOVA was used to calculate the statistical difference. Differences with *p* < 0.05 were considered statistically significant.

## 3. Results

### 3.1. In Vitro Antigastric Cancer Effects of CBD

To study the effect of CBD on gastric cancer in vitro, we treated the gastric cancer SGC-7901 cells with different concentrations of CBD for 24 and 48 h. The activity of SGC-7901 cells after CBD treatment was detected by the CCK-8 assay. The results showed that CBD (5–40 μg/mL) significantly inhibited SGC-7901 cell proliferation in a concentration-dependent manner, with an IC_50_ value of 23.4 μg/mL after 24 h of treatment (Figure 1a). The inhibition rate was significantly higher in SGC-7901 cells after 48 h of treatment with CBD than after 24 h of treatment (Figure 1a). In addition, the cell colony formation assay demonstrated that CBD effectively inhibited colony formation in SGC-7901 cells at concentrations of 10, 20, and 40 μg/mL (Figure 1b,c).

### 3.2. CBD Induces Cell Cycle Arrest at the G0–G1 Phase in SGC-7901 Cells

To further investigate the inhibitory effect of CBD on the proliferation of SGC-7901 cells, we examined the cell cycle distribution of SGC-7901 cells after 24 h of treatment with CBD. Flow cytometry analysis showed that the proportion of SGC-7901 cells that were at the G0–G1 phase significantly increased after treatment with CBD compared to the control group (Figure 2a,b). It should be noted that the CBD-induced G0–G1 phase arrest was obviously released in SGC 7901 cells after 24 h of incubation with fresh culture media, as evidenced by no significant difference in the proportion of SGC-7901 cells at the G0–G1 phase being found between the 20-μg/mL CBD treatment group and the control group (Figure 2c,d). When the G1/S cycle is retarded, the content of the CDK2/cyclin E complex is correspondingly reduced [30]. Therefore, we examined the expression levels of CDK2 and cyclin E in cells after CBD treatment, which were significantly lower than the control group (Figure 2e,f). These results indicated that CBD could effectively induce cell cycle arrest at the G0–G1 phase by inhibiting CDK2 and cyclin E expression.

### 3.3. Effects of CBD on the ATM/p53/p21 Signaling Pathway

ATM is activated during DNA damage, which in turn can upregulate the expression level of P21, downregulate the expression level of P53 [31], and subsequently inhibit the formation of CDK2/cyclin E complexes [32]. Previous experiments have shown that CBD can induce cell cycle arrest and decrease the expression of CDK2/cyclin E in SGC-7901 cells. Therefore, we examined the protein expression levels of ATM, p53, and p21 in SGC-7901 cells after CBD treatment and found that both ATM and p21 expression were elevated, whereas that of the p53 protein decreased (Figure 3a,b).

### 3.4. CBD Promotes Apoptosis in SGC-7901 Cells

To explore whether the observed inhibition of the proliferation effect of CBD is related to cell apoptosis, Hoechst 33258 staining of nuclei was performed. Fluorescence microscopy revealed that there were more abnormal nuclei in the CBD administration group compared to the control. These nuclei had distinct apoptotic features such as nuclear contraction, irregular condensation of chromatin, and apoptotic bodies (Figure 4a). We also analyzed the apoptotic rate of CBD-treated SGC-7901 cells using an Annexin V FITC/PI double staining kit and flow cytometry. We found that as the concentration of CBD increased, the percentage of apoptotic cells in the SGC-7901 population increased (Figure 4b,c). These results indicated that CBD effectively induced apoptosis in SGC-7901 cells.

### 3.5. Effect of CBD on Apoptosis-Related Proteins

In the above experiments, we observed that CBD could effectively induce apoptosis in SGC-7901 cells. To explore the intrinsic mechanism of this effect, we performed western blot analysis to identify the core proteins associated with apoptosis. The results showed that CBD significantly reduced the expression of caspase-3 and caspase-9 at the same time and markedly increased the content of cleaved caspase-3 and cleaved caspase-9 (Figure 5a,b). We also investigated the proteins that activate the caspase family of proteins, such as cytochrome C and Apaf-1, by western blotting. The results showed that CBD remarkably increased cytoplasm cytochrome C and Apaf-1 protein expression levels in SGC-7901 cells (Figure 5c,d). These results implied that CBD induced SGC-7901 cell apoptosis through a mitochondrial-dependent apoptosis pathway.

### 3.6. The Effect of CBD on the Mitochondrial Apoptosis Signaling Pathway

Next, the levels of mitochondrial apoptosis-related proteins in CBD-treated SGC-7901 cells were examined by western blotting. We found that CBD significantly increased the protein expression levels of Bad and Bax and decreased that of Bcl-2 relative to the control group (Figure 6a,b). We also further examined the mitochondrial membrane potential of CBD-treated SGC-7901 cells using JC-1 staining. Flow cytometry analysis showed that the ratio of red to green fluorescence significantly decreased in CBD-treated SGC-7901 cell groups compared to the control group (Figure 6c,d). Fluorescence microscopy revealed a significant increase in green fluorescence intensity and a decrease in red fluorescence intensity in CBD-treated SGC-7901 cells (Figure 6e). The results confirmed that CBD induced SGC-7901 cell apoptosis via the mitochondrial-dependent apoptosis pathway.

### 3.7. The Effect of CBD on the ROS Levels of SGC-7901 Cells

Based on the antioxidant activity of CBD, we investigated the changes in ROS levels in CBD-treated SGC-7901 cells using a DCFH-DA fluorescence probe. The DCFH-DA probe is nonfluorescent in its initial form, but it can be easily oxidized by intracellular ROS, leading to the formation of fluorescent product dichlorofluorescein (DCF) [33]. After 2′,7′-dichlorofluorescein diacetate (DCFH-DA) staining, flow cytometry analysis showed that the intensity of green fluorescence (DCF) was markedly enhanced in CBD-induced SGC-7901 cells compared to the control (Figure 7a,b).

## 4. Discussion

CBD is the main chemical in and a nonaddictive component of the medical plant cannabis [34]. Studies have shown that CBD can inhibit tumor cell proliferation and metastasis or induce autophagy or apoptosis [35] in various cancers such as glioma [36], leukemia [37], prostate cancer [38], and breast cancer [39]. In this study, we found that CBD could arrest SGC-7901 cells at the G0–G1 phase and induce apoptosis of SGC-7901 cells by activating the mitochondrial apoptosis pathway.

Checkpoints are important regulatory nodes of the cell cycle. Cells can only enter the next cell cycle after passing these checkpoints [40]. The function of the G0–G1 phase detection point is to integrate and transmit complex intracellular and extracellular signals, such as various growth factors, mitogens, and DNA damage, as well as to determine whether cells are undergoing division and apoptosis [41]. When the cells begin to synthesize DNA during G1–S conversion, CDK2 will combine with its regulatory subunit cyclin E to form a Cdk2/Cyclin E complex, leading to Rb phosphorylation, and then E2F factor is released and cells are accelerated into the S phase [42]. During G0–G1 cycle arrest, the content of the CDK2/cyclin E complex decreases accordingly [43]. Our data showed that the protein expression levels of CDK2 and cyclin E in the SGC-7901 cells decreased after CBD treatment, which in turn reduced the formation of CDK2/cyclin E complexes, ultimately arresting SGC-7901 cells at the G0–G1 phase.

The cell cycle refers to the entire process of continuous cell division. When cell cycle arrest occurs during cell division, it is often due to damage or errors that are difficult to repair during cell division [44]. This theory suggests that CBD-induced G1 arrest may be due to DNA damage that is difficult to repair in cells. At the same time, intracellular DNA is damaged, ATM and ATR (ATM and Rad 3-related) are the centers of the stress response, and ATM/ATR signaling is activated [45]. The ATM/ATR signaling pathway can repair damaged DNA by modulating the activity of various proteins. At present, it is generally believed that the ATM/ATR signaling pathway mediates G0–G1 arrest by regulating the expression of p53. ATM can directly regulate p53, which increases the p53 protein level, which in turn enhances p21 transcription [46]. Activation of ATM during DNA damage can upregulate the expression of the p21 protein and downregulate p53 protein expression. Finally, the formation of the CDK2/cyclin E complexes is inhibited, and the cell cycle is arrested at the G0–G1 phase. The results of this study are consistent with the above theory. After treatment with CBD, ATM protein expression levels increased, and the ATM protein was activated in SGC-7901 cells. Meanwhile, CBD increased the expression of p21 and downregulated the expression of p53 in SGC-7901 cells, which in turn led to cell cycle arrest at the G0–G1 phase.

Apoptosis is a programmed cell death of the body’s cells [47]. A recent study showed that CBD can induce apoptosis in breast cancer cells [48]. This study found that the levels of cleaved caspase-3 and -9 were upregulated in SGC-7901 cells after 24 h of treatment with CBD, subsequently inducing cell apoptosis. Considering that the mitochondria-mediated caspase-dependent pathway is a major apoptotic pathway, we then examined the levels of anti- and pro-apoptotic Bcl-2 family proteins in the mitochondria-dependent apoptotic pathway. We found that CBD upregulated Bax and downregulated Bcl-2 in CBD-treated SGC-7901 cells, leading to a decrease in the ratio of Bcl-2/Bax.

Mitochondrial membrane permeability increases and mitochondrial transmembrane potential decreases when the ratio of Bcl-2/Bax decreases [49]. Large amounts of the apoptotic initiator mitochondrial cytochrome C then flow into the cytoplasm. Cytochrome C in the cytoplasm activates Apaf-1 and caspase-9 and -3, which cleaves DNA and produces apoptotic bodies that ultimately lead to apoptosis [50]. In our study, we further examined the expression levels of the aforementioned proteins in the mitochondrial apoptotic signaling pathway. Our data showed that CBD significantly decreased the mitochondrial transmembrane potential, released mitochondrial cytochrome C into the cytoplasm, and activated Apaf-1 and caspase-9 and -3, which ultimately resulted in SGC-7901 cell apoptosis.

Previous studies have demonstrated that the continuous increase of intracellular ROS leads to DNA damage [51,52]. After the “sensing” or “detection” of DNA damage, the ATM/ATR signal is activated, which subsequently upregulates p21 and downregulates p53, leading to cell cycle arrest at the G0–G1 phase by decreasing the level of the CDK2/cyclin E complexes [53]. Moreover, the continuous increase in intracellular ROS levels leads to the continuous opening of mitochondrial permeability transition pore (mPTP), which reduces the mitochondrial transmembrane potential [54], and the release of cytochrome C into the cytoplasm, which results in a decrease in mitochondrial membrane potential and induces cell apoptosis through the mitochondrial-dependent pathway [55]. Considering the increase in the ROS levels in Jurkat leukemia EL-4 cells induced by CBD [22], we examined the ROS levels in CBD-treated SGC-7901 cells and found that the intracellular ROS levels in SGC-7901 cells significantly increased after CBD treatment. Taken together, these results indicated that CDB-induced cell cycle arrest and cell apoptosis of SGC-7901 cells were associated with the increasing intracellular ROS levels.

However, some limitations to the antitumor effects of CBD in this study should be noted. First, the antigastric cancer effect of CBD was only examined in one human gastric cancer cell line, SGC-7901, and thus other gastric cancer cells (e.g., human gastric cancer BGC-823 cells and mouse gastric cancer MFC cells) should be employed to further explore the therapeutic effects of CBD. Second, the change in ROS levels was examined in CBD-treated SGC-7901 cells, so some antioxidants could be employed to evaluate the role of ROS in CBD-induced apoptosis of SGC 7901 cells. In addition, the cytotoxicity of CBD in normal cells was not determined. However, we searched for reports of the cytotoxicity of CBD in normal cells. It has been reported that no significant effects on physiological parameters (heart rate, blood pressure, and body temperature) and psychological functions are found after CBD treatment, and high doses up to 1500 mg/day of CBD are tolerated well in humans [26], implying that CBD has no obvious cytotoxicity in normal tissues/cells.

In conclusion, our results indicated that CBD upregulated CDK2 and cyclin E in SGC-7901 cells by increasing ATM expression, which in turn induced cell cycle arrest at the G0–G1 phase. CBD also upregulated Bax and Bad and downregulated Bcl-2 in SGC-7901 cells, which in turn activated the mitochondrial-dependent apoptotic pathway and ultimately induced apoptosis. These findings may be utilized in the development of CBD as a potential drug for the treatment of gastric cancer.

## Figures and Tables

**Figure 1 biomolecules-09-00302-f001:**
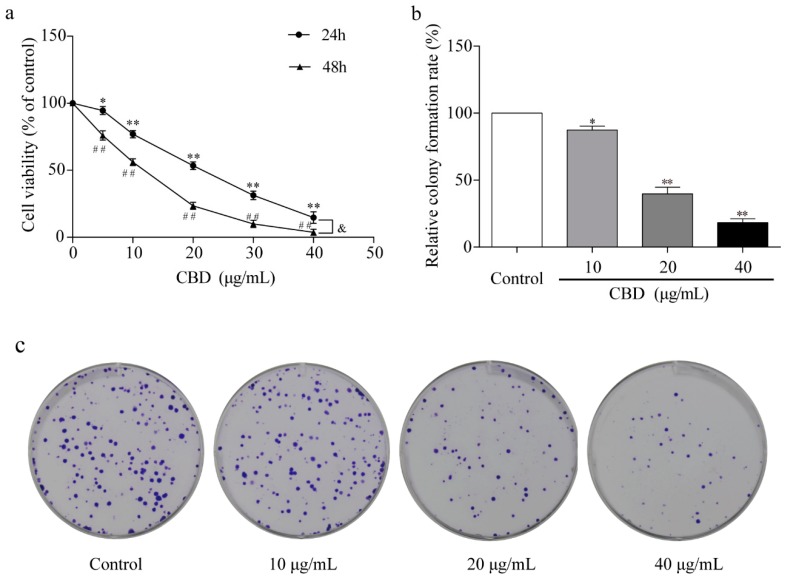
Inhibition of proliferation of SGC-7901 cells by cannabidiol (CBD). (**a**) SGC-7901 cells were seeded into 96-well plates, and cells were treated with different concentrations of CBD for 24 and 48 h. The survival rate of cells treated with CBD was measured by the CCK-8 method. (**b**) Statistical analysis of cell colony formation rate in CBD-treated SGC-7901 cells. (**c**) Representative images of cell colonies in CBD-treated SGC-7901 cells. All data are expressed as the mean ± SD of three independent experiments. * *p* < 0.05, ** *p* < 0.01 compared to the control (24 h). ^##^
*p* < 0.01 compared to the control (48 h). ^&^
*p* < 0.05 compared to the 24-h CBD-treated cells.

**Figure 2 biomolecules-09-00302-f002:**
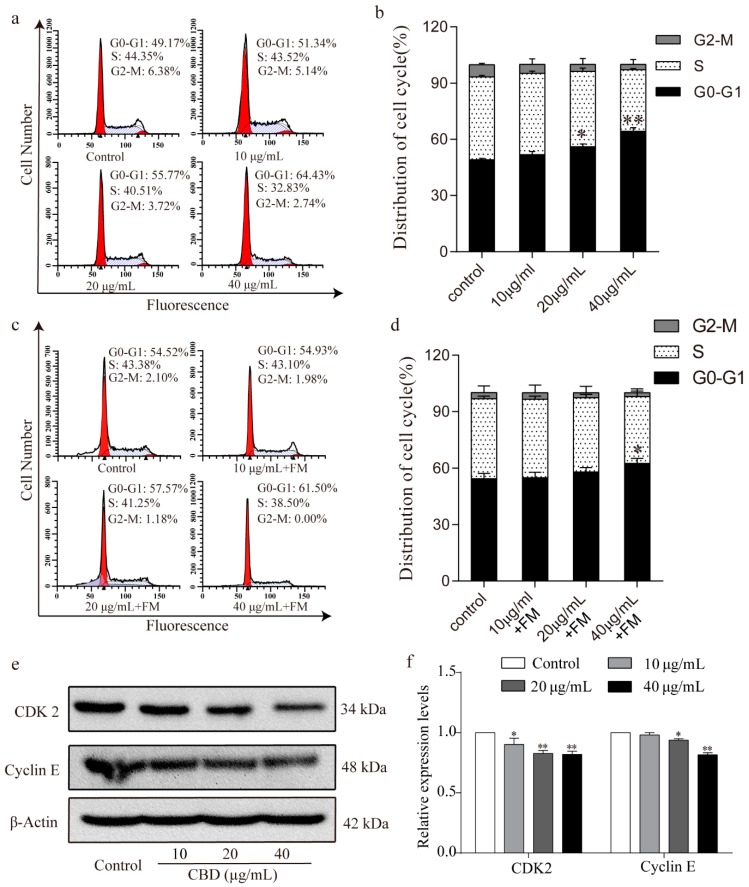
CBD induced G0–G1 cell cycle arrest of SGC-7901 cells. (**a**) Flow cytometry was used to determine the cell cycle distribution of CBD-treated SGC-7901 cells. (**b**) Statistical analysis of the cell cycle distribution of SGC-7901 cells treated by CBD. (**c**) SGC-7901 cells were treated with CBD for 24 h, followed by 24 h of incubation with fresh culture media (FM), and then the cycle distribution was examined by flow cytometry. (**d**) Statistical analysis of the cell cycle distribution of SGC-7901 cells after 24 h of CBD treatment and 24 h of incubation with fresh culture media (FM). (**e**) The expression levels of CDK2 and cyclin E protein in CBD-treated SGC-7901 cells were detected by western blotting. (**f**) Statistical analysis of the levels of CDK2 and cyclin E in CBD-treated SGC-7901 cells. The ratio of protein levels was normalized according to the values of the control. All data are expressed as the mean ± SD of three independent experiments. * *p* < 0.05, ** *p* < 0.01 compared to the control.

**Figure 3 biomolecules-09-00302-f003:**
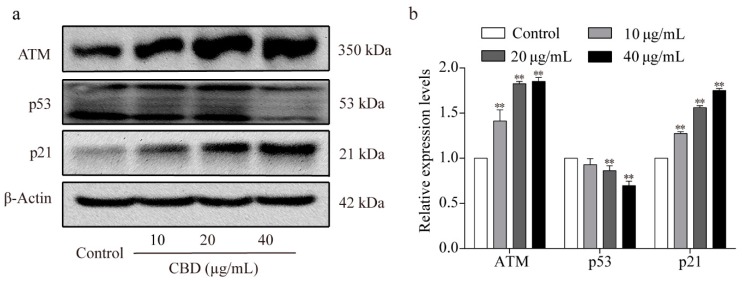
Effect of CBD on ATM/P53/P21 signaling protein levels. (**a**) Western blotting was used to detect the protein expression levels of ATM, p53, and p21 in SGC-7901 cells after 24 h of treatment with CBD. (**b**) Statistical analysis of the levels of ATM, p53, and p21 in SGC-7901 cells treated with CBD. The ratio of protein levels was normalized according to the values of the control. All data are expressed as the mean ± SD of three independent experiments. * *p* < 0.05, ** *p* < 0.01 compared to the control.

**Figure 4 biomolecules-09-00302-f004:**
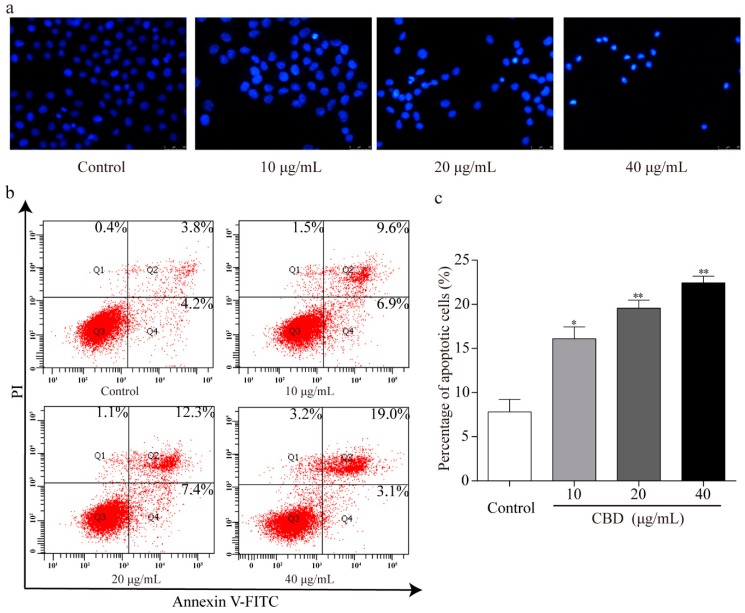
CBD could effectively induce apoptosis in SGC-7901 cells. (**a**) After 24 h of treatment with CBD, SGC-7901 cells showed typical morphological changes after staining with Hoechst 33258. (**b**) After 24 h of treatment with CBD, the apoptosis rate of SGC-7901 cells was detected by Annexin V-FITC/PI double staining. (**c**) Statistical analysis of the apoptotic rate of the SGC-7901 cell population after CBD treatment. All data are expressed as the mean ± SD of three independent experiments. * *p* < 0.05, ** *p* < 0.01 compared to the control.

**Figure 5 biomolecules-09-00302-f005:**
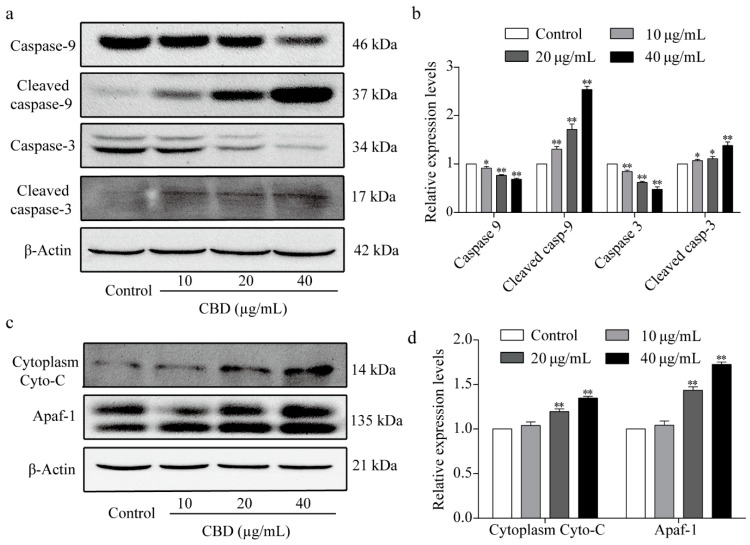
Effect of CBD on the expression of apoptosis-related proteins. (**a**) After 24 h of treatment with CBD, the protein expression levels of caspase-9, cleaved caspase-9, caspase-3, and cleaved caspase-3 in SGC-7901 cells were determined by western blotting. (**b**) Statistical analysis of protein relative levels of caspase-9, cleaved caspase-9, caspase-3, and cleaved caspase-3 in SGC-7901 cells treated with CBD. The ratio of protein levels was normalized according to the values of the control. (**c**) After 24 h of treatment with CBD, the levels of cytoplasm cytochrome C and Apaf-1 in SGC-7901 cells were determined by western blotting. (**d**) Statistical analysis of the levels of cytoplasm cytochrome C and Apaf-1 in CBD-treated SGC-7901 cells. The ratio of protein levels was normalized according to the values of the control. All data are expressed as the mean ± SD of three independent experiments. * *p* < 0.05, ** *p* < 0.01 compared to the control.

**Figure 6 biomolecules-09-00302-f006:**
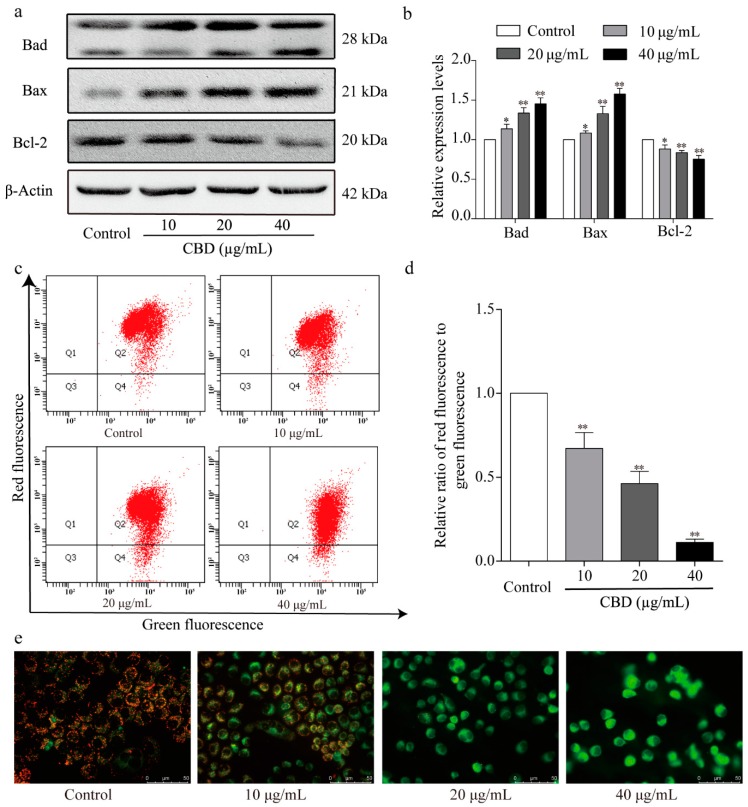
Effect of CBD on the expression of signaling molecules in the mitochondrial apoptotic pathway. (**a**) After 24 h of treatment with CBD, the protein levels of Bad, Bax, and Bcl-2 in CBD-treated SGC-7901 cells were determined by western blotting. (**b**) Statistical analysis of relative protein expression levels of Bad, Bax, and Bcl-2 in CBD-treated SGC-7901 cells. The ratio of the proteins was normalized according to the values of the control. (**c**) After 24 h of treatment with CBD, the changes in the mitochondrial membrane potential in SGC-7901 cells were detected using a JC-1 (CBIC2(3)) mitochondrial membrane potential assay kit. (**d**) Quantitative analysis of the mitochondrial membrane potential of SGC-7901 cells. (**e**) After 24 h of treatment with CBD, SGC-7901 cells were stained with JC-1 dye, and the fluorescence distribution was observed under a fluorescence microscope. All data are expressed as the mean ± SD of three independent experiments. * *p* < 0.05, ** *p* < 0.01 compared to the control.

**Figure 7 biomolecules-09-00302-f007:**
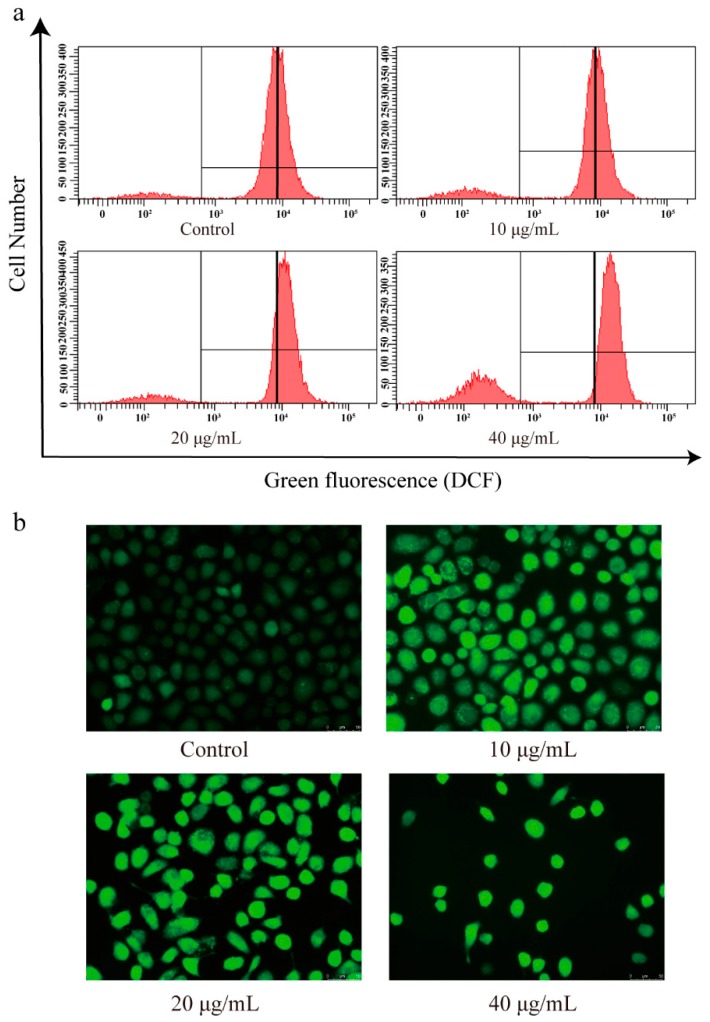
Effect of CBD on reactive oxygen species (ROS) levels in SGC-7901 cells. (**a**) After 24 h of treatment with CBD, the changes in reactive oxygen species (ROS) content in SGC-7901 cells were determined using an ROS assay kit. (**b**) After staining with an ROS species assay kit, the fluorescence distribution in CBD-treated SGC-7901 cells was observed under a fluorescence microscope.

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
