# Peer review of "Cannabidiol Induces Cell Cycle Arrest and Cell Apoptosis in Human Gastric Cancer SGC-7901 Cells"

_biomolecules, 2019, doi:10.3390/biom9080302_

Round 1

Reviewer 1 Report

In this study authors demonstrated that CBD inhibited proliferation of SGC-7901 cell line by inducing apoptosis and cell cycle arrest. 

Data are very interesting and promising, but there is no healthy control. Authors should demonstrate that the same CBD concentrations have no effect on cell viability of healthy cells to exclude toxicity and demonstrate that CBD acts specifically on tumor cells. A single test would be sufficient, for example apoptosis on healthy cells.

Change Figure 2b: statistycal analysis of cell cycle is more clear like Fig. 7 of Lamorte et al. "Future in the Past: Azorella glabra Wedd. as a Source of New Natural Compounds with Antiproliferative and Cytotoxic Activity on Multiple Myeloma Cells." Int J Mol Sci. 2018 Oct 26;19(11).

Insert this paper in references.  

Author Response

Point 1: In this study authors demonstrated that CBD inhibited proliferation of SGC-7901 cell line by inducing apoptosis and cell cycle arrest.

Response 1: Thanks for the comments.

Point 2: Data are very interesting and promising, but there is no healthy control. Authors should demonstrate that the same CBD concentrations have no effect on cell viability of healthy cells to exclude toxicity and demonstrate that CBD acts specifically on tumor cells. A single test would be sufficient, for example apoptosis on healthy cells.

Response 2: Thanks for the comments. As the reviewer’s concern, the safety concern of CBD should be noted. The anti-tumor effects of CBD were examined in human gastric cancer SGC-7901 cells, but the cytotoxicity of CBD in normal cells wasn’t determined. To release the reviewer’s concern, we searched for the reports for the cytotoxicity of CBD in normal cells. It has been reported that no significant effect in physiological parameters (heart rate, blood pressure and body temperature) and psychological functions was found after CBD treatment, and high doses up to 1,500 mg/day of CBD are well-tolerated in humans (Bergamaschi MM, et al., Curr Drug Saf. 2011), implying that CBD has no obvious cytotoxicity in normal tissues/cells. We have added this report in the Discussion section. Please refer to line 423-428 in the updated manuscript.

Point 3: Change Figure 2b: statistical analysis of cell cycle is more clear like Fig. 7 of Lamorte et al. "Future in the Past: Azorella glabra Wedd. as a Source of New Natural Compounds with Antiproliferative and Cytotoxic Activity on Multiple Myeloma Cells." Int J Mol Sci. 2018 Oct 26;19(11). Insert this paper in references.

Response 3: Thanks for the comments. According to the reviewer’s suggestion, we have resupplied the Figure 2b and added the above paper in the References section. Please refer to line 129 in the updated manuscript.

Reviewer 2 Report

Zhang et al. in the manuscript “Cannabidiol induces cell cycle arrest and cell apoptosis in human gastric cancer SGC-7901 cells” presents the results on antiproliferative and apoptosis inducing activity of non-additive component of Cannabis sativa L., cannabidiol (CBD), against SGC 7901 tumour cells. The data presented are interesting and support the future antitumor studies on CBD and its derivatives. However the manuscript needs some improvement.

Introduction section:

The references of Ramer & Hinz, Adv Pharmacol, 2017 (PMID: 28826542) and Bergamaschi et al., Curr Drug Saf, 2011 (PMID: 22129319) should be cited and discussed within the content of CBD anticancer activities and safety.

Methods section:

This part needs substantial editing. There are numerous misspellings and grammar errors (especially TENSES).

Just few examples of grammatical mistakes:

Lane 123 – “boiled at 37°C”. Shall be “incubated at 37°C”.

Lane 133 – “culture was cultured”. Better is “culture was incubated”.

Lane 142 – “method was measured according”. How about “was performed according”.

Lane 221 – “thrice”. Did you mean twice?

Results section:

- There is a discrepancy between CKK-8 method description and results presented in Fig.1a. In the methods you mentioned that CBD was tested at 10, 20 and 40 ug/ml, but Fig.1a shows the 5, 10, 20, 30 and 40 ug/ml of CDB.

- Please supplement the Fig.2 with results of cell cycle profiles of cells released from the G0/G1 block induced by CBD treatment (24 h CBD exposure followed by 24 h incubation in fresh media; block-release). It will be essential to analyse and show, if cells treated with CBD could be released from the cell cycle arrest or do they die (result of the colony formation assay suggests that cells are not able to recover after exposure to 20 ug/ml CBD). Optional: Include the data on cell cycle profile of SGC 7901 cells treated with 20 ug/ml CBD for 6, 12, 24, 36 and 48 h. It is obvious that some cells after treatment with CBD died (detected markers of apoptosis), but what happen to cells that survived CBD treatment, and how long it takes them to recover is unclear.

- Note in the legend of Fig.3,4,5 for how long were cells treated with CBD.

- Explain in the text of the results section (Fig.7) how change in the fluorescence intensity correlates with the observed level of ROS production in CBD treated cells.

- To confirm the role of ROS in CBD-induced apoptosis of SGC 7901 tumour cells, please include the results of cells pre-treated with some antioxidants (e.g. NAC) followed by CBD treatment. If ROS play important role than you expect to restore cell viability, relieve oxidative stress and block the apoptotic effects of CBD treatment.

Discussion section:

Lane 369: correct “…which increases p53 protein proteins that…”

Author Response

Point 1: Zhang et al. in the manuscript “Cannabidiol induces cell cycle arrest and cell apoptosis in human gastric cancer SGC-7901 cells” presents the results on antiproliferative and apoptosis inducing activity of non-additive component of Cannabis sativa L., cannabidiol (CBD), against SGC 7901 tumor cells. The data presented are interesting and support the future antitumor studies on CBD and its derivatives. However, the manuscript needs some improvement.

Response 1: Thanks for the comments. We had tried our best to address the comments by careful revision using point-by-point fashion to improve the quality of our manuscript. Please refer to the revisions as below.

Point 2: Introduction section: The references of Ramer & Hinz, Adv Pharmacol, 2017 (PMID: 28826542) and Bergamaschi et al., Curr Drug Saf, 2011 (PMID: 22129319) should be cited and discussed within the content of CBD anticancer activities and safety.

Response 2: Thanks for the suggestions. We have added these references in the Introduction section. Please refer to line 56-58 and 65-66 in the updated manuscript.

Point 3: This part needs substantial editing. There are numerous misspellings and grammar errors (especially TENSES). Just few examples of grammatical mistakes:

Lane 123 – “boiled at 37°C”. Shall be “incubated at 37°C”.

Lane 133 – “culture was cultured”. Better is “culture was incubated”.

Lane 142 – “method was measured according”. How about “was performed according”.

Lane 221 – “thrice”. Did you mean twice?

Response 3: Thanks for the kindly reminder. We have corrected the above errors. Please refer to line 125, 136, 145 and 225 in the updated manuscript.

Point 4: Results section: There is a discrepancy between CKK-8 method description and results presented in Fig.1a. In the methods you mentioned that CBD was tested at 10, 20 and 40 ug/ml, but Fig.1a shows the 5, 10, 20, 30 and 40 ug/ml of CDB.

Response 4: Thanks for the kindly reminder. We have corrected the typing error. Please refer to line 88 in the updated manuscript.

Point 5: -Please supplement the Fig.2 with results of cell cycle profiles of cells released from the G0/G1 block induced by CBD treatment (24 h CBD exposure followed by 24 h incubation in fresh media; block-release). It will be essential to analyze and show, if cells treated with CBD could be released from the cell cycle arrest or do they die (result of the colony formation assay suggests that cells are not able to recover after exposure to 20 μg/ml CBD). Optional: Include the data on cell cycle profile of SGC 7901 cells treated with 20 μg/ml CBD for 6, 12, 24, 36 and 48 h. It is obvious that some cells after treatment with CBD died (detected markers of apoptosis), but what happen to cells that survived after CBD treatment, and how long it takes them to recover is unclear.

Response 5: Thanks for the valuable suggestions. According the reviewer’s comments, we examined the cell cycle distribution of SGC 7901 cells after 24 h CBD treatment and 24 h incubation with fresh culture media. We found that the CBD-induced G0-G1 phase arrest was obviously released in SGC 7901 cells after 24 h incubation with fresh culture media, as evidenced by no significant difference in the proportion of SGC-7901 cells at G0-G1 phase was found between 20 μg/ml CBD treatment group and the control group. However, our results also indicated that CBD effectively induced SGC-7901 cell apoptosis, and ultimately lead to the inhibition of cell colony formation after exposure to 20 μg/ml CBD. We have added this result in the Results section. Please refer to Figure 2c-d, line 121-122, 251-255 and 263-266 in the updated manuscript.

Point 6: Note in the legend of Fig.3,4,5 for how long were cells treated with CBD.

Response 6: Thanks for the kindly reminder. We have added the treatment times in the legend of Fig.3,4,5. Please refer to line 280-281, 296-298, 313-314 and 318 in the updated manuscript.

Point 7: Explain in the text of the results section (Fig.7) how change in the fluorescence intensity correlates with the observed level of ROS production in CBD treated cells.

Response 7: Thanks for the comments. In this experiment, the ROS levels in CBD-treated SGC-7901 cells were examined using DCFH-DA fluorescence probe. The probe DCFH-DA is non-fluorescent in the initial form, but it can be easily oxidized by intracellular ROS, leading to the formation of fluorescent product dichlorofluorescein (DCF) (Rastogi RP, et al., Biochem Biophys Res Commun., 2010). We have added the explanation in the Results section. Please refer to line 346-349 in the updated manuscript.

Point 8: To confirm the role of ROS in CBD-induced apoptosis of SGC 7901 tumour cells, please include the results of cells pre-treated with some antioxidants (e.g. NAC) followed by CBD treatment. If ROS play important role than you expect to restore cell viability, relieve oxidative stress and block the apoptotic effects of CBD treatment.

Response 8: Thanks for the valuable suggestions. In this study, we found that the intracellular ROS levels in SGC-7901 cells significantly increased after CBD treatment, so we supposed that CDB could induce cell apoptosis of SGC-7901 cells by increasing intracellular ROS levels. As the reviewer’s comments, some antioxidants could be employed to evaluate the role of ROS in CBD-induced apoptosis of SGC 7901 cells. To release the reviewer’s concern, we re-depicted the conclusion in the Discussion section as follow: these results indicated that CDB-induced cell cycle arrest and cell apoptosis of SGC-7901 cells are associated with the increasing intracellular ROS levels. Please refer to line 416-417 in the updated manuscript. In addition, we have mentioned the limitation in discussion. Please refer to Discussion (line 244-247) in the updated manuscript. Please refer to line 421-423 in the updated manuscript.

Point 9: Discussion section: Lane 369: correct “…which increases p53 protein proteins that…”

Response 9: Thanks for the kindly reminder. We have corrected the typing error. Please refer to line 381 in the updated manuscript.

Round 2

Reviewer 2 Report

No further recommendations.